# Spatial aggregation choice in the era of digital and administrative surveillance data

**Elizabeth C. Lee**[1], **Ali Arab**[2], **Vittoria Colizza**[3], **Shweta Bansal**[4]*

**1** Department of Epidemiology, Johns Hopkins Bloomberg School of Public Health, Baltimore, Maryland, United States of America, **2** Department of Mathematics and Statistics, Georgetown University, Washington, District of Columbia, United States of America, **3** INSERM, Sorbonne Université, Institut Pierre Louis d'Epidémiologie et de Santé Publique, Paris, France, **4** Department of Biology, Georgetown University, Washington, District of Columbia, United States of America

* shweta.bansal@georgetown.edu

## Abstract

Traditional disease surveillance is increasingly being complemented by data from non-traditional sources like medical claims, electronic health records, and participatory syndromic data platforms. As non-traditional data are often collected at the individual-level and are convenience samples from a population, choices must be made on the aggregation of these data for epidemiological inference. Our study seeks to understand the influence of spatial aggregation choice on our understanding of disease spread with a case study of influenza-like illness in the United States. Using U.S. medical claims data from 2002 to 2009, we examined the epidemic source location, onset and peak season timing, and epidemic duration of influenza seasons for data aggregated to the county and state scales. We also compared spatial autocorrelation and tested the relative magnitude of spatial aggregation differences between onset and peak measures of disease burden. We found discrepancies in the inferred epidemic source locations and estimated influenza season onsets and peaks when comparing county and state-level data. Spatial autocorrelation was detected across more expansive geographic ranges during the peak season as compared to the early flu season, and there were greater spatial aggregation differences in early season measures as well. Epidemiological inferences are more sensitive to spatial scale early on during U.S. influenza seasons, when there is greater heterogeneity in timing, intensity, and geographic spread of the epidemics. Users of non-traditional disease surveillance should carefully consider how to extract accurate disease signals from finer-scaled data for early use in disease outbreaks.

## Author summary

Administrative health records, social media streams like Twitter, and participatory surveillance systems like Influenzanet are increasingly available for infectious disease surveillance, but are often geographically aggregated to preserve data privacy and confidentiality. We explored how an arbitrary choice in the spatial aggregation of non-traditional disease data sources may influence estimates of disease burden and epidemiological

**Data Availability Statement:** The medical claims database is not publicly available; they were obtained from IMS Health, now IQVIA, which may be contacted at https://www.iqvia.com/. All other model input data and map base layers are made publicly available by the US Census Bureau. Model

output data and code are available at https://github.com/bansallab/spatialaggregation.

**Funding:** Research reported in this publication was supported by the Jayne Koskinas Ted Giovanis Foundation for Health and Policy (http://jktgfoundation.org/); and the National Institute of General Medical Sciences of the National Institutes of Health under award number R01GM123007. The content is solely the responsibility of the authors and does not necessarily represent the official views of the National Institutes of Health, and the funders had no role in study design, data collection and analysis, decision to publish, or preparation of the manuscript.

**Competing interests:** The authors have declared that no competing interests exist.

understanding of an outbreak. Using influenza-like illness as measured through a medical claims database as our case study, we find that there is substantial variation in influenza season timing and magnitude across spatial scales due to which spatial aggregation could lead to misleading estimates of epidemiological quantities. In particular, we find that epidemiological inferences are more sensitive to spatial scale early on during U.S. influenza seasons, when there is greater heterogeneity in timing, intensity, and geographic spread of the epidemics. Non-traditional disease surveillance may have distinct advantages in reporting speed and volume, but care is required when aggregating this data for spatial epidemiological analysis.

## Introduction

Effective disease surveillance systems seek to capture accurate, representative, and timely disease data in the face of complex logistical challenges and limited human resources [1]. As these data are typically collected at centralized locations like sentinel healthcare facilities and summarized according to political administrative boundaries, there are natural spatial units that may be incorporated into the surveillance system design and reporting. Aggregating surveillance data to administrative boundaries is useful because these units are used in the allocation and distribution of resources and the development of public health guidelines.

While the hope is that spatial and temporal heterogeneity in reported surveillance corresponds to the true underlying disease burden, biases in measurement may contribute to inaccurate estimates. One potential source of bias when working with aggregated surveillance data, often overlooked, stems from choices in the design and aggregation of the reporting data stream itself. In disease ecology, it is well-documented that ecological processes are sensitive to spatial scale, that differences in scale may explain seemingly-conflicting data, and that disease distributions are the result of hierarchical processes that occur on different scales [e.g. [2–6]]. Parallel concerns arise in spatial statistics, where the ecological and atomistic fallacies warn against the extension of statistical conclusions from populations to individuals and vice versa [e.g. [7–9]]. In epidemiology, statistical methods that account for the hierarchical nature of spatial data have been developed to improve disease mapping for small area aggregated health data [e.g. [10, 11]].

Non-traditional disease data such as digital data streams, syndromic disease reporting, and medical claims were not necessarily generated for the purpose of disease surveillance, but they have the potential to provide information relevant to disease tracking in a timely and cost-efficient way across large geographic scales [12–17]. Traditional surveillance systems are designed to meet pre-determined objectives such as routine surveillance or outbreak detection, for a fixed set of syndromes or diseases in a specific population. Non-traditional data are typically more voluminous and collected at the individual level, but they often capture a convenience sample limited by user biases. For example, medical claims data captures only individuals with health insurance, while Twitter users with a specific geolocation tag may be younger than the general population in that location. Moreover, the collection of non-traditional disease data is not often designed with attention to logistical reporting constraints. Consequently, epidemiologists and policy makers increasingly have new choices in how to aggregate these records spatially and temporally. Noise and random variability may mask epidemiologically-relevant disease signals in data at finer spatial and temporal scales, and we have limited understanding about how these choices might affect subsequent inference [18–20].

Using U.S. medical claims data for influenza-like illness as a case study, we consider the issue of 'spatial aggregation choice' among potentially novel sources of surveillance data. First we characterize influenza season dynamics from 2002–2003 through 2008–2009 across different spatial aggregation scales. We examine defining influenza season features such as the epidemic source location, onset and peak season timing, and epidemic duration with data aggregated to the county and state levels. Finally, we compare spatial autocorrelation for burden between the early and peak influenza seasons, and test the relative magnitude of spatial aggregation differences for seasonal measures related to timing and intensity. This work highlights the scenarios under which spatial aggregation choice are important, particularly when considering the use of alternative surveillance data streams.

## Methods

### Medical claims data

Weekly visits for influenza-like illness (ILI) and any diagnosis from October 2002 to April 2009 were obtained from a records-level database of U.S. medical claims managed by IMS Health and processed to the county scale. ILI was defined with International Classification of Diseases, Ninth Revision (ICD-9) codes for: direct mention of influenza, fever combined with respiratory symptoms or febrile viral illness, or prescription of oseltamivir, while any diagnosis visits represent all possible medical diagnoses including ILI (also see [21]). We also obtained metadata from IMS Health on the percentage of reporting physicians and the estimated effective physician coverage by visit volume [21]. Over the years in our study period, our medical claims database represented an average of 24% of visits for any diagnosis from 37% of all health care providers across 95% of U.S. counties during influenza season months [21].

We also aggregated visits for ILI and any diagnosis to the U.S. state- and region-levels, where region boundaries were defined by the groupings of states by the U.S. Department of Health and Human Services.

We performed the same data processing procedure for each county-, state- and region-level time series of ILI per any diagnosis visits (ILI ratio) that has been described elsewhere [21]. In brief, ILI intensity is calculated as a detrended ILI ratio during the flu period from November through March. The flu period is defined as the maximum consecutive period when the ILI ratio exceeds an epidemic threshold (minimum of at least two weeks).

### Defining disease burden and spatial aggregation difference

The study considered five measures of influenza disease burden—two measures of timing (onset and peak flu season timing), two measures of intensity (onset and peak intensity), and epidemic duration—at county, state, and region scales. In the below definitions, the *intensity* of influenza activity in a given location and time refers to the time series of the detrended ILI ratio (See details about the intensity calculation at [21]).

We defined *onset timing* as the number of weeks from week number 40 (first week of October) until the first week in the epidemic period. We defined *peak timing* as the number of weeks from week 40 until the week with the maximum epidemic intensity during the epidemic period. The *epidemic duration* was the number of weeks where the ILI intensity exceeded the epidemic threshold.

Proxies for prevalence during the onset flu season and peak flu season were calculated like relative risks; the *onset intensity* and *peak intensity* for a given county, state, or region was defined as its risk relative to a single, national 'expected' onset and peak prevalence, respectively. This 'expected prevalence,' calculated for each influenza season, was the county population-weighted mean of the associated intensity measure. The onset flu season was identified as

a 2–3 week flu season period with the greatest exponential growth rate, while the peak flu season was identified as the week with the maximum ILI intensity.

We defined *spatial aggregation difference* as the difference between a given influenza disease burden measure at an aggregated spatial scale (i.e., state or region) and the county spatial scale (e.g., $\mu_{state} - \mu_{county}$, where $\mu$ is a burden measure). As burden measures are normalized, they may be compared across spatial scales and the scale of the spatial aggregation difference is the same as that of each individual burden measure. A positive spatial aggregation difference indicates that state- or region-level data over-represented disease burden magnitude (onset and peak intensity) or had later epidemic timing (onset or peak timing) relative to county measures. Among timing measures, spatial aggregation error of 20 means that state surveillance data presented epidemic onset or epidemic peak 20 weeks after county surveillance data. Among intensity measures, a spatial aggregation error of -1 means that state surveillance data reported $e^{-1} \approx 0.37$ times the risk of county surveillance data.

## Inferring probable source location

Using seasonal time series of intensity, we identified the top 10% of locations (at the county or state scale) with the earliest epidemic onset for each season as potential source locations and calculated the Euclidean distances between the centroids of potential source locations and all other locations. We then used the Pearson correlation coefficient ($H_o$: no difference from zero) between distance to potential source location and onset week to identify probable county or state source locations for a given influenza season (higher correlation coefficient means higher probability of being source location).

## Examining spatial dependence in influenza disease burden

We plotted spatial correlograms to examine the global spatial autocorrelation of the four county-level summary measures of disease burden in the statistical programming language R with the `ncf` package [22]. A two-sided permutation test was performed with 500 permutations to identify correlations that deviated significantly from zero ($H_o$: no difference from zero).

## Comparing spatial aggregation differences across measures and scales

We tested whether spatial aggregation difference was greater among early season or peak season measures of disease burden, and whether state- or region-level aggregations generated greater differences across all measures of disease burden. To compare onset and peak season measures, we paired the spatial aggregation differences for county-season observations across all influenza seasons within our study period for 1) onset timing and peak timing and 2) onset intensity and peak intensity, and tests were performed for both state- and region-level values. To compare differences among state- or region-level aggregations, we paired state-county and region-county differences by county observation for each of the four disease burden measures.

We compared spatial aggregation difference with Bayesian intercept models (effectively, a Bayesian paired t-test) that accounted for county spatial dependence (See SM Methods). The models were implemented with approximate Bayesian inference in R using Integrated Nested Laplace Approximations (INLA) with the `INLA` package (www.r-inla.org) [23, 24].

Positive estimates mean that 1) spatial aggregation differences for peak timing are greater than those for onset timing, 2) spatial aggregation differences for peak intensity are greater than those for early intensity, or 3) spatial aggregation differences for region and county are greater than those for state and county, and vice versa for negative values. If the 95% credible intervals for $\beta_0$ fail to overlap with zero, we interpret that there is a statistically significant

difference between the measures contributing to $\delta_i$. We used relatively non-informative normal priors for $\beta_0$ and log-gamma priors for the precision term $\tau_\phi$.

## Results

We explore the scales of influenza surveillance using county-level U.S. medical claims data representing 2.5 billion visits from upwards of 120,000 health care providers each year for influenza seasons from 2002–2003 through 2008–2009. There was evident heterogeneity in the intensity and timing of ILI activity between counties and their aggregated state and HHS region scales (Fig 1).

### Probable epidemic source locations rarely overlap between county- and state-level data streams

We inferred the most probable epidemic counties and states independently for each influenza season. Across all seasons, we found disagreement in the top two most probable source states and the top 50 most probable source counties (Fig 2). Probable source counties partially overlapped with probable source states only in a few influenza seasons and in a small set of locations: four counties representing 41% of the population of Rhode Island overlapped in the 2004–2005 season; nine counties in California (33% of state population) and seven counties in Nevada (21% of state population) overlapped in the 2005–2006 season; eight counties in Alabama (7% of state population) and 28 counties in Georgia (6% of state population) overlapped in the 2006–2007 season.

### A majority of county data streams achieve onset and peak timing milestones before state data streams

To elucidate the discrepancy between county and state epidemic source locations, we compared the influenza season onset and peak week between county and state scales. While ILI spread was sometimes very rapid, with influenza season onset striking almost all counties within a given state at once, these patterns were not consistent across seasons or states (Fig O in S1 Text).

State-level flu season onset and peak timing tended to occur after the majority of counties in the state had already achieved those milestones. Across the 2002–2003 through 2008–2009 influenza seasons, a mean of 62% and 70% of state populations had already experienced the onset and peak of the influenza season by the times when the aggregated state-level data achieved its influenza season onset and peak, respectively (Fig 3). County population size did not appear to have an association with onset or peak timing (Fig A-N in S1 Text).

Through visual examination of correlograms, we found that spatial autocorrelation remained present at greater distances for peak measures than early season measures of disease burden, suggesting that seasonal dynamics become more spatially synchronized as the influenza season progresses (Fig 4). Autocorrelation declined to zero at 1177 km and 1359 km for onset and peak timing and at 809 km and 1140 km for onset and peak intensity, respectively.

While county-level epidemics had greater heterogeneity in epidemic duration, often with longer right-skewed tails, epidemic durations were similar across spatial scales (Fig P in S1 Text). There was greater variability in epidemic duration between influenza seasons than between different spatial scales. Only in the HHS region centered in New York did the distributions in epidemic duration appear to be shifted. However, this region may be particularly subject to discrepancies related to spatial scales because it represents the smallest geographic area in the study region.

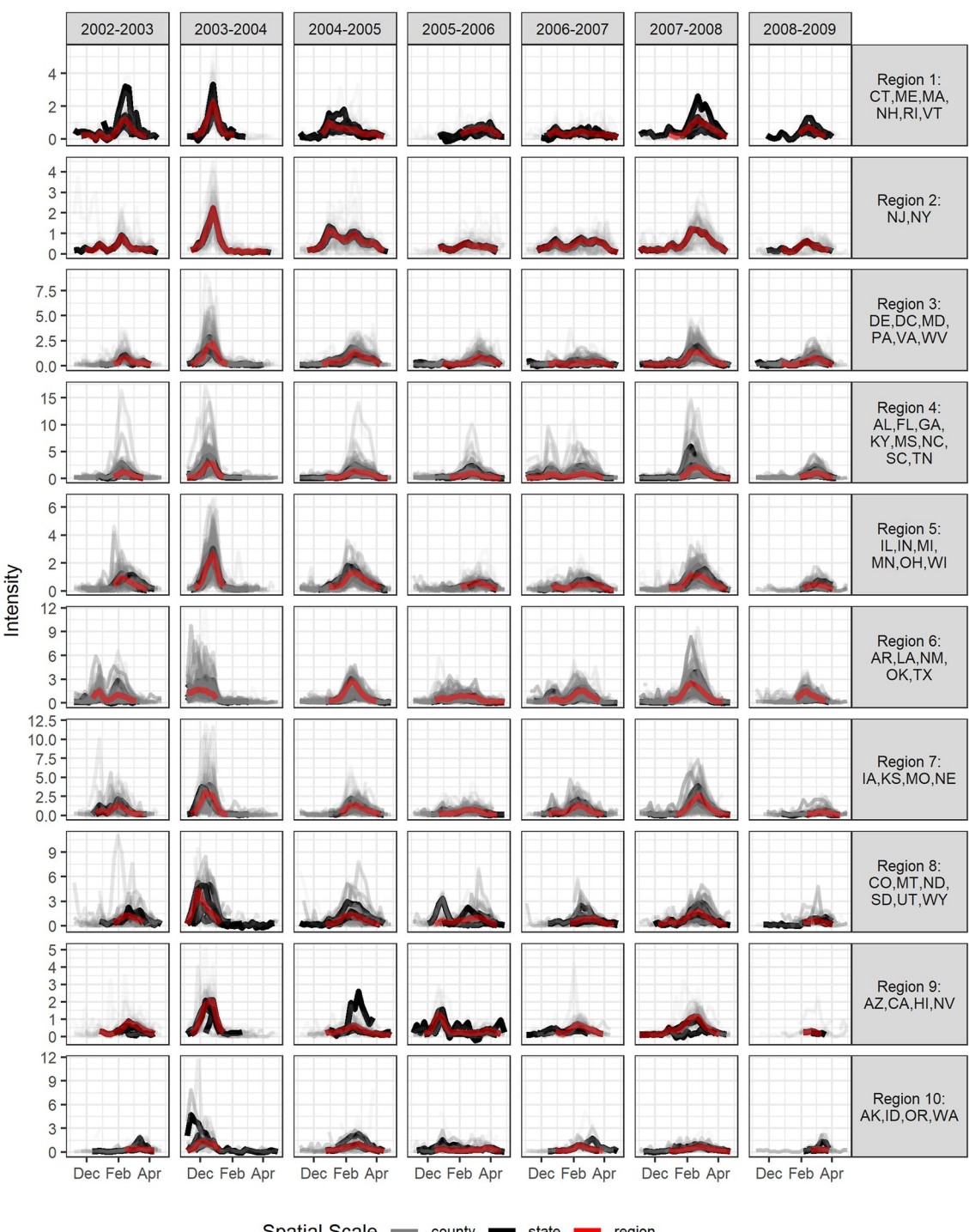

**Fig 1. ILI intensity by influenza season from 2002–2003 through 2008–2009 across 10 HHS regions.** ILI intensity is displayed for all available counties and states in a given HHS region in different colors (grey for counties, black for states, and red for region). Some regions (such as Region 1) have fewer counties than others so heterogeneity at the county level may be less apparent.

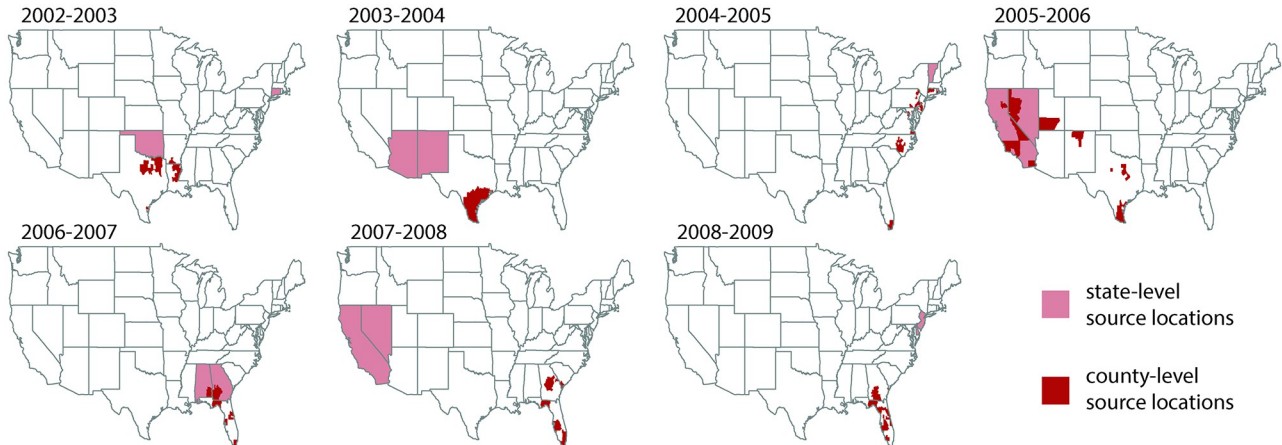

**Fig 2. Most probable influenza season U.S. source locations at state and county scales across all influenza seasons.** We present the two states (pink) and 50 counties (red) that are the most probable source locations for each influenza season from 2002–2003 through 2008–2009. Probable source counties were partially contained within probable source states only in the 2004–2005, 2005–2006, and 2006–2007 influenza seasons. When inferring probable source locations, disagreement between county- and state-level analyses was common. The map base layer is from the US Census Bureau.

County-level maps of burden and *spatial aggregation difference* for an example influenza season for onset timing, peak timing, onset intensity, and peak are displayed in the supplement (Fig Q in S1 Text).

## Spatial aggregation differences are more prevalent at epidemic onset than at peak flu season

We compared spatial aggregation differences between onset and peak timing and between onset and peak intensity using a Bayesian procedure that may be viewed as a paired t-test for spatially correlated data. The estimates indicate that spatial aggregation differences between state and county measures were greater for onset timing than peak timing and for onset intensity than peak intensity (Table 1). This means that there was greater heterogeneity in the timing and intensity of early season measures than in the peak season measures. Region-county differences were also greater for onset intensity than peak intensity (Table A in S1 Text).

Region-county differences were larger than state-county ones for timing measures, while state-county differences were larger than region-county ones for measures of disease intensity (Table B in S1 Text).

## Discussion

Administrative health records, social media streams like Twitter, and participatory surveillance systems like Influenzanet, Flu Near You, and Facebook COVID-19 Symptom Survey are increasingly available for disease surveillance, but use of these data for epidemiological analysis is subject to 'spatial aggregation choice' [15, 17]. In this study, we examined how an arbitrary choice in the spatial aggregation of non-traditional disease data sources may influence estimates of disease burden and epidemiological understanding of an outbreak. First, we describe the dynamics and burden of influenza-like illness across the United States from 2002–2003 through 2008–2009 with medical claims data across the county, state, and HHS region spatial scales. We observed substantial heterogeneity in influenza season timing and magnitude across spatial scales and found that analyses performed with county-level and state-level data could provide contradictory results regarding inference on the most probable epidemic source

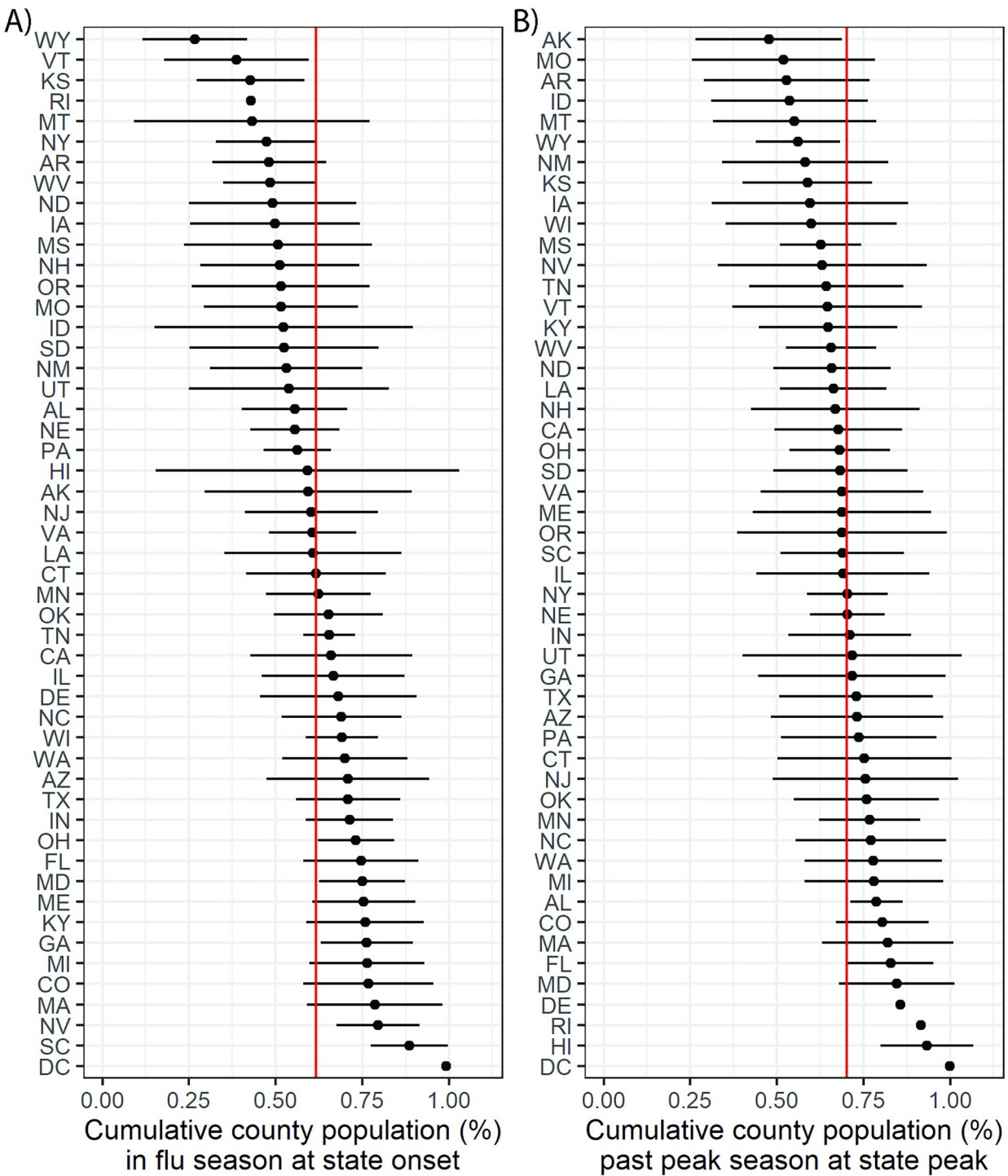

**Fig 3. Comparison of county and state influenza season onset and peak timing.** We present the cumulative percentage of county populations that have experienced (A) influenza season onset and (B) the influenza season peak by the time that these milestones have been achieved by the aggregated state-level data. For each state abbreviation (rows), the point represents the mean across influenza seasons from 2002–2003 through 2008–2009 while the horizontal line indicates the range of one standard deviation on either side of the mean. The red vertical lines indicate the mean of the mean values across states.

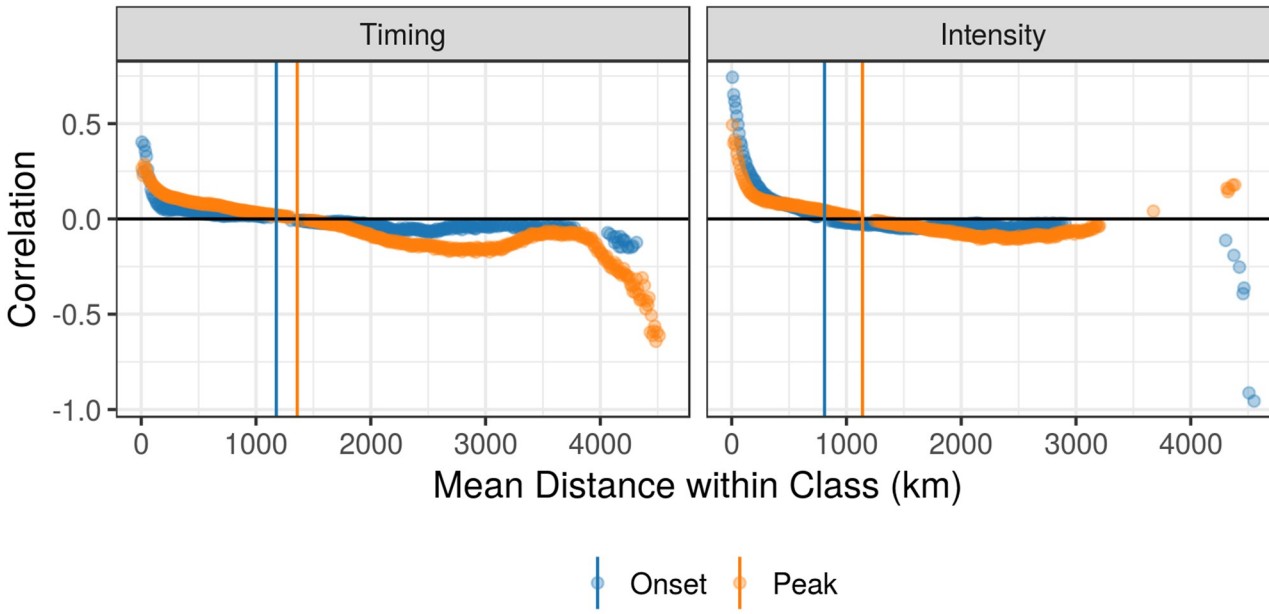

**Fig 4. Spatial correlograms for timing and intensity measures across all influenza seasons.** We present spatial autocorrelation among counties within specified distance classes for timing measures (left) and intensity measures (right). Early season measures (onset timing and early season intensity) are represented in blue and peak season measures (peak timing and intensity) are in orange. Points are displayed only if the p-value for a two-sided permutation test to evaluate correlation is less than 0.01. Colored vertical lines indicate the mean distance where county measures are no more similar than that expected by chance in a given region.

location. State-level timing measures provided delayed information about the onset and peak season timings, and timing-related measures had greater spatial heterogeneity in disease burden and spatial aggregation error than did intensity-related measures.

We initially hypothesized that influenza epidemics aggregated to larger spatial scales would have longer epidemic duration than county-level data because a state-level or region-level epidemic should represent the set of all lower-level epidemics, which are staggered in time. During our study period, however, state-level onset and peak season timings occurred only after 60–70% of the state's population had experienced those milestones (as reported by county-level data), and epidemic duration was similar across spatial scales. Our analysis suggests that spatial aggregation makes influenza season outbreaks less sensitive to onset and peak timing detection. The identification of source locations was also highly scale-dependent; probable source counties were only occasionally located within the probable source state, and when there was overlap, those counties represented only a small proportion of the state population. We also highlight that seasons with more geographically synchronized flu epidemics, which can occur in antigenically novel or severe seasons [25, 26], are not any more likely to have overlap in source locations across spatial scales. We thus hypothesize that source locations are independent of peak dynamics.

**Table 1. Comparison of state-county spatial aggregation differences between onset and peak season measures.** Negative values mean that spatial aggregation estimates for peak measures were smaller than spatial aggregation differences for onset measures. Bolded values denote mean estimates that we interpret to have statistical significance; that is, the 95% credible intervals did not overlap with zero.

| State-County Comparison | Estimate (95%CI) |
|---|---|
| Peak-Onset Timing | **-0.23** (-0.29, -0.16) |
| Peak-Early Intensity | **-0.31** (-0.33, -0.30) |

These results suggest that spatially-aggregated data are less reliable in representing early season dynamics. This may be because the timing and intensity of ILI activity appears to be more heterogeneous in the early season than the peak season [27], and heterogeneity is associated with greater spatial aggregation differences (Fig T-W in S1 Text). Counties were spatially autocorrelated at greater distances for both peak timing and peak intensity as compared to onset timing and early season intensity, respectively (Fig 4), and spatial aggregation differences were smaller for peak measures than early season measures (Table 1 and Table A in S1 Text). Two factors may contribute to these differences between early and peak season: 1) there are less reliable disease signals during the early flu season, and 2) observations in epidemic onset are asynchronized, but they become more spatially synchronized as the season progresses. Together, these results bolster the hypothesis that seasonal influenza is seeded to many locations and spread primarily through local transmission [28], while prior work suggests that school-holiday-associated contact reductions may play a role in synchronizing influenza outbreaks [29].

Our study suggests that spatial aggregation choice is most critical in early influenza season surveillance (i.e., identifying source locations and early season inference), particularly for assessing season onset (Fig R-S in S1 Text). Delayed detection of season onset and inaccurate estimation of early season intensity may lessen agility of policymakers and healthcare facilities to anticipate staffing and hospital supply needs as they prepare for the peak influenza season activity. Nevertheless, further work should be done to verify the generalizability of our results to different disease syndromes and data sources. Our conclusions about when and how spatial aggregation choice is most important may be conflated with other data reporting processes, such as the expanding geographic coverage of our medical claims data over time, the distribution of reporting healthcare facilities, variability in reporting quality (driven by differences in healthcare and surveillance resources), clustered use of certain ICD-9 diagnosis codes (driven by hospital practices or knowledge-sharing between physicians, for example), and healthcare access, as well as the stochastic variation in influenza season dynamics itself. In addition, changes of mobility over time (e.g. due to family or touristic travels during the winter holidays) associated with changes of contact patterns (e.g. due to closure of schools during holidays) are known to contribute to influenza diffusion [29, 30] and may therefore affect the spatial aggregation.

As big data becomes more prevalent and fine-scale targeting and measurement becomes the norm in infectious disease surveillance, spatial aggregation and zoning biases, discrepancies between statistical inference when the boundaries of contiguous spatial units are rearranged [31, 32], may become regular concerns for epidemiologists. This case study has a direct link to the modifiable areal unit problem (MAUP), a phenomenon which describes how spatial aggregation of data can yield different statistical results and highlights the need for sensitivity analyses examining spatial scale [32]. While we sought to describe differences in influenza season features across spatial scales, other recent work pursues the identification of epidemiology-driven geographic regions as a potential solution to this problem [25]. At this juncture, where traditional, administrative, and digital data may be used in disease surveillance, it is critical to develop general methodologies that can extract useful disease signals from fine-scaled data early on in an outbreak [16].

## Supporting information

**S1 Text. Detailed descriptions about data processing, methodological choices, sensitivity analyses, and supporting evidence.** The Supporting Information includes Table A and B and Figures A to W.
(PDF)

## Acknowledgments

This work was made possible by a data agreement between IQVIA and the RAPIDD Program of the Science & Technology Directorate, Department of Homeland Security and the Fogarty International Center, National Institutes of Health.

## Author Contributions

**Conceptualization:** Elizabeth C. Lee, Shweta Bansal.

**Data curation:** Elizabeth C. Lee.

**Formal analysis:** Elizabeth C. Lee, Shweta Bansal.

**Funding acquisition:** Elizabeth C. Lee, Shweta Bansal.

**Investigation:** Elizabeth C. Lee, Vittoria Colizza.

**Methodology:** Elizabeth C. Lee, Ali Arab, Vittoria Colizza, Shweta Bansal.

**Supervision:** Ali Arab, Vittoria Colizza, Shweta Bansal.

**Validation:** Elizabeth C. Lee, Ali Arab, Vittoria Colizza.

**Visualization:** Elizabeth C. Lee.

**Writing – original draft:** Elizabeth C. Lee.

**Writing – review & editing:** Elizabeth C. Lee, Ali Arab, Vittoria Colizza, Shweta Bansal.

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
