## [Decision Letter · Decision Letter 0]

28 Jan 2022

PDIG-D-21-00097

Spatial aggregation choice in the era of digital and administrative surveillance data

PLOS Digital Health

Dear Dr. Lee,

Thank you for submitting your manuscript to PLOS Digital Health. After careful consideration, we feel that it has merit but does not fully meet PLOS Digital Health's publication criteria as it currently stands. Therefore, we invite you to submit a revised version of the manuscript that addresses the points raised during the review process.

We look forward to receiving your revised manuscript.

Kind regards,

Yuan Lai, Ph.D.

Academic Editor

PLOS Digital Health

Journal Requirements:

1. Please update your Competing Interests statement. If you have no competing interests to declare, please state: “The authors have declared that no competing interests exist.”

2. We ask that a manuscript source file is provided at Revision. Please upload your manuscript file as a .doc, .docx, .rtf or .tex. If you are providing a .tex file, please upload it under the item type ‘LaTeX Source File’ and leave your .pdf version as the item type ‘Manuscript’.

3. Please provide separate figure files in .tif or .eps format only and remove any figures embedded in your manuscript file. Please ensure that all files are under our size limit of 20MB.

For more information about how to convert your figure files please see our guidelines: https://journals.plos.org/digitalhealth/s/figures

4. Please provide us with a direct link to the base layer of the map used in Figure 2, Figure S17, Figure S18, Figure S19, and ensure this location is also included in the figure legend. 

Please note that, because all PLOS articles are published under a CC BY license (creativecommons.org/licenses/by/4.0/), we cannot publish proprietary maps such as Google Maps, Mapquest or other copyrighted maps. If your map was obtained from a copyrighted source please amend the figure so that the base map used is from an openly available source.

Please note that only the following CC BY licences are compatible with PLOS licence: CC BY 4.0, CC BY 2.0 and CC BY 3.0, meanwhile such licences as CC BY-ND 3.0 and others are not compatible due to additional restrictions. If you are unsure whether you can use a map or not, please do reach out and we will be able to help you. 

The following websites are good examples of where you can source open access or public domain maps:

Additional Editor Comments (if provided):

Dear authors, we have received all reviewers' decisions, and please refer to their comments below for the revision. In particular, both Reviewer #1 and #3 suggest better explanations on some definitions and terminology. Please address these feedbacks accordingly. Many thanks for your submission and apologize for the rather long reviewing process.

Reviewers' comments:

Reviewer's Responses to Questions

**Comments to the Author**

1. Does this manuscript meet PLOS Digital Health’s publication criteria? Is the manuscript technically sound, and do the data support the conclusions? The manuscript must describe methodologically and ethically rigorous research with conclusions that are appropriately drawn based on the data presented.

Reviewer #1: Yes

Reviewer #2: Yes

Reviewer #3: Yes

2. Has the statistical analysis been performed appropriately and rigorously?

Reviewer #1: I don't know

Reviewer #2: Yes

Reviewer #3: Yes

3. Have the authors made all data underlying the findings in their manuscript fully available (please refer to the Data Availability Statement at the start of the manuscript PDF file)?

Reviewer #1: Yes

Reviewer #2: Yes

Reviewer #3: No

4. Is the manuscript presented in an intelligible fashion and written in standard English?

Reviewer #1: Yes

Reviewer #2: Yes

Reviewer #3: Yes

5. Review Comments to the Author

Reviewer #1: This paper describes how spatial aggregation (region, state, and county level) can influence measures of disease onset and peak while using non-traditional surveillance data sources. Using ILI and medical claims data as a use case, this work shows how differences in early season measures are observed across different spatial aggregations. In my opinion, these findings contribute to disease surveillance and epidemiological work, especially regarding non-traditional data sources. To the extent of my knowledge, the methods are appropriate for assessing this research question. The study findings, presentation of the work, and rigor of the methods all contribute to my recommendation to publish with minor revisions. 

The overall paper provides a clear narrative about how spatial aggregation choice may impact epidemiological inference of non-traditional data sources. While there are no major concerns on my part regarding the study, there are some places in the paper that could benefit from refined language clarity, and enrichment. These areas are detailed below.

In the introduction, the wording around “political administrative boundaries” in the first paragraph needs addressing. Perhaps there is a word missing. The introduction also mentions that “methods that account for the hierarchical nature of spatial data have been developed to improve disease mapping and the study of disease dynamics” ahead of a paragraph that details that this does not necessarily apply to non-traditional disease data. I recommend clarifying the statement to show it is more applicable to traditional epi data, or that this work needs to be done for non-tradition. I also recommend reviewing the text to remove the same descriptive word more than once in a statement, such as often in ““Non-traditional data are more voluminous and often collected at the individual level, but they often capture a convenience sample limited by user biases.”

For the methods, I think adding what definition of ILI was used (IMS health definition, CSTE, etc.) would be helpful detail. Elaborating on the definitions of weekly ILI visits and diagnoses would also be helpful. Are these diagnoses of ILI, flu, etc. None of the enrichments need to be lengthy or exhaustive, just more informative. I am interested in how expected prevalence was calculated with population weights yet did not vary across counties. Some enrichment around this would be interesting, perhaps in the discussion. Additionally, where the time series of season intensity described in “Inferring probable source location” different than the time series sued to defined intensity? For the Euclidean distances, it is the only measurement in its section that is only county, not state or county. How, if any, where distances state to county handled? Lastly, clearly stating the four county-level summary measures of disease burden after the foist use of this language would be a helpful detail to readers, providing more clarity as they navigate the paper. 

In the results, I think you should specify the following is across all observed seasons: “We found disagreement in the top two most probable source states and the top 50 most probable source counties (Figure 2)”. In the sub section “A majority of county data streams achieve onset and peak timing milestones before state data streams”, please address the repeating word (achieved) in the sentence beginning with “State-level flu season onset and peak timing tended”. Additionally, I suggest a slight re-phrase so this does not read exactly like methods section: “We defined spatial aggregation difference as the difference between a give influenza disease burden measure at aggregated spatial scales”. For Figure 2, the county data for HHS region 1 is more difficult to see than the other HHS regions. Highlighting the difference in scale for intensity across regions in the caption would be informative. While the figure’s focus is intraregional comparison across spatial scale, it might be good to note the differences. For Table 1, is the text starting with “The two negative estimates indicate” part of the title? Should that be moved to the caption? Or explained in the text describing results in the figure? 

For the discussion, I would emphasize that the goal of this study is more focused on influence of spatial aggregation, especially as novel data streams become available for influenza. I would include the goals around spatial aggregation in this sentence as it was the focus, rather than the current phrasing. Some additional considerations to enrich this section include additional biases specific to medical claims data that could relate to the results seen in spatial aggregation (beyond geographic coverage). This could include access to care, facility aggregation (particularly in rural areas), etc. Similarly, it may be interesting to briefly address how mobility may play into differential results of spatial aggregation in early and peak seasons. Lastly, detailing why onset and peak location and timing are important to capture arcuately would be a good addition. 

The findings of this study, especially about the plurality of state populations experiencing ILI onset ahead of state-level data indicating such, are very interesting. I think Figure 2 did a nice job of displaying the onset location differences across county and state level data. ILI is a good use case for this work and it is helpful to see how non-traditional and digital data may be influenced differently by spatial choice than traditional epi data, particularly as more of these data for respiratory illnesses emerge in light of COVID-19.

Reviewer #2: Because this manuscript is about the spatial autocorrelation, I think it should be also showed the cold-spot, not only the hot-spot areas. It can be really helpful for the readers and also the policy makers in order to find the comparison between the areas. Also, it can boots the improvement in this research in the future. Perhaps using multivariate approach, in order to find the major or other causes of diseases spread.

I think it will be really important in order to boost the similar research in other countries, especially in my country, Indonesia, in order to boost the surveillance ILI data and spatial utilization in public health sector.

In page 7, I found a technical issue in the paragraph

I think the authors should explain the limitation of the study. In manuscript, it is not explained well for me.

Reviewer #3: The study makes good contribution to the current knowledge. Spatial aggregation differences are critical to statistical inference out of multi-level data. However, the measurement of some key variables in the study looks not easy to follow. For example, " the onset intensity and peak intensity metrics were defined as risks relative to the ‘expected’ onset and peak prevalence...". It would be difficult for readers to learn how the intensity was exactly measured. What are the "risks"? How "expected" was defined? AND the key variable in this study, spatial aggregation difference, was defined as "the difference between a given influenza disease burden measure...." It was just one sentence measuring this. The authors should elaborate more with this measurement. And they should make it easy to read. How burden is related to spatial aggregation difference?

6. PLOS authors have the option to publish the peer review history of their article (what does this mean?). If published, this will include your full peer review and any attached files.

**Do you want your identity to be public for this peer review?** For information about this choice, including consent withdrawal, please see our Privacy Policy.

Reviewer #1: No

Reviewer #2: Yes: Dhihram Tenrisau

Reviewer #3: Yes: Ge Zhan

---

## [Decision Letter · Decision Letter 1]

11 Apr 2022

Spatial aggregation choice in the era of digital and administrative surveillance data

PDIG-D-21-00097R1

Dear Dr. Lee,

We are pleased to inform you that your manuscript 'Spatial aggregation choice in the era of digital and administrative surveillance data' has been provisionally accepted for publication in PLOS Digital Health.

Best regards,

Yuan Lai, Ph.D.

Academic Editor

PLOS Digital Health

Dear Dr. Lee,

It is my pleasure to inform that we received all feedbacks from the reviewers regarding you revised manuscript "Spatial aggregation choice in the era of digital and administrative surveillance data".

Please see reviewers' decision and comments below.

Best,

Yuan

Reviewer Comments (if any, and for reference):

Reviewer's Responses to Questions

**Comments to the Author**

1. If the authors have adequately addressed your comments raised in a previous round of review and you feel that this manuscript is now acceptable for publication, you may indicate that here to bypass the “Comments to the Author” section, enter your conflict of interest statement in the “Confidential to Editor” section, and submit your "Accept" recommendation.

Reviewer #1: All comments have been addressed

Reviewer #2: All comments have been addressed

Reviewer #3: All comments have been addressed

2. Does this manuscript meet PLOS Digital Health’s publication criteria? Is the manuscript technically sound, and do the data support the conclusions? The manuscript must describe methodologically and ethically rigorous research with conclusions that are appropriately drawn based on the data presented.

Reviewer #1: Yes

Reviewer #2: Yes

Reviewer #3: Yes

3. Has the statistical analysis been performed appropriately and rigorously?

Reviewer #1: Yes

Reviewer #2: Yes

Reviewer #3: Yes

4. Have the authors made all data underlying the findings in their manuscript fully available (please refer to the Data Availability Statement at the start of the manuscript PDF file)?

Reviewer #1: Yes

Reviewer #2: Yes

Reviewer #3: Yes

5. Is the manuscript presented in an intelligible fashion and written in standard English?

Reviewer #1: Yes

Reviewer #2: Yes

Reviewer #3: Yes

6. Review Comments to the Author

Reviewer #1: The authors addressed all of my comments from the initial review. Overall, I am pleased with the updated version of the manuscript.

Reviewer #2: Thank you for your answers. I hope this research can give the impact for the surveillance research for the future. If you don't mind can you also add the github link or rmarkdown? I think it can be really important to the reader, so they can use and learn your methods in his R application. Thank you

Reviewer #3: Comments have been well adressed in the revision.

7. PLOS authors have the option to publish the peer review history of their article (what does this mean?). If published, this will include your full peer review and any attached files.

**Do you want your identity to be public for this peer review?** For information about this choice, including consent withdrawal, please see our Privacy Policy.

Reviewer #1: No

Reviewer #2: No

Reviewer #3: **Yes: **Ge Zhan
